# Determining the Potential of DNA Damage Response (DDR) Inhibitors in Cervical Cancer Therapy

**DOI:** 10.3390/cancers14174288

**Published:** 2022-09-01

**Authors:** Santu Saha, Stuart Rundle, Ioannis C. Kotsopoulos, Jacob Begbie, Rachel Howarth, Isabel Y. Pappworth, Asima Mukhopadhyay, Ali Kucukmetin, Kevin J. Marchbank, Nicola Curtin

**Affiliations:** 1Translational and Clinical Research Institute, Faculty of Medical Sciences, Newcastle University, Framlington Place, Newcastle upon Tyne NE2 4HH, UK or; 2The Northern Gynaecological Oncology Centre (NGOC), Queen Elizabeth Hospital, Gateshead NE9 6SX, UK; 3University College London Hospitals NHS Foundation Trust, 250 Euston Rd, London NW1 2PG, UK; 4Addenbrooke’s Hospital, Cambridge CB2 0QQ, UK; 5Translational and Clinical Research Institute, National Renal Complement Therapeutics Centre, Newcastle University, Newcastle upon Tyne NE2 4HH, UK; 6Kolkata Gynecological Oncology Trials and Translational Research Group, Chittaranjan National Cancer Institute, Kolkata 700026, India; 7Department of Gynaecological Oncology, James Cook University Hospital, Middlesbrough TS4 3BW, UK; 8Faculty of Medical Sciences, Newcastle University, Newcastle upon Tyne NE2 4HH, UK; 9Translational and Clinical Research Institute, Newcastle University, Newcastle upon Tyne NE2 4HH, UK

**Keywords:** cervical cancer, cisplatin, radiotherapy, DDR inhibitors, kidney toxicity

## Abstract

**Simple Summary:**

Cervical cancer (CC) is the fourth most common cause of cancer deaths in women. For patients where surgery is not an option, they are treated with cisplatin and radiotherapy (RT) that acts by damaging DNA. However, response is poor, and cisplatin causes kidney injury. The DNA damage response (DDR) consists of coordinated action of proteins that signal DNA damage to stop cell proliferation and promote repair. In this study, we show four different DDR inhibitors targeting PARP, ATR, CHK1 and WEE1 that kill CC cells. These inhibitors also increase the ability of RT and cisplatin to kill CC cells to varying degrees. Additionally, we show that cisplatin-induced kidney injury is due to over-activation of PARP and can be reduced by co-administration of a PARP inhibitor. DDR inhibition is therefore a promising strategy to both increase the effectiveness of current treatment and to protect kidneys from cisplatin-induced toxicity.

**Abstract:**

Cisplatin-based chemo-radiotherapy (CRT) is the standard treatment for advanced cervical cancer (CC) but the response rate is poor (46–72%) and cisplatin is nephrotoxic. Therefore, better treatment of CC is urgently needed. We have directly compared, for the first time, the cytotoxicity of four DDR inhibitors (rucaparib/PARPi, VE-821/ATRi, PF-477736/CHK1i and MK-1775/WEE1i) as single agents, and in combination with cisplatin and radiotherapy (RT) in a panel of CC cells. All inhibitors alone caused concentration-dependent cytotoxicity. Low ATM and DNA-PKcs levels were associated with greater VE-821 cytotoxicity. Cisplatin induced ATR, CHK1 and WEE1 activity in all of the cell lines. Cisplatin only activated PARP in S-phase cells, but RT activated PARP in the entire population. Rucaparib was the most potent radiosensitiser and VE-821 was the most potent chemosensitiser. VE-821, PF-47736 and MK-1775 attenuated cisplatin-induced S-phase arrest but tended to increase G2 phase accumulation. In mice, cisplatin-induced acute kidney injury was associated with oxidative stress and PARP activation and was prevented by rucaparib. Therefore, while all inhibitors investigated may increase the efficacy of CRT, the greatest clinical potential of rucaparib may be in limiting kidney damage, which is dose-limiting.

## 1. Introduction

Cervical cancer (CC) is the fourth most frequent malignancy accounting for 7.7% cancer mortality in women [1]. Persistent infection by high-risk human papilloma virus (HR-HPV) is the cause for the vast majority of CC. In the UK, screening and HPV immunisation has resulted in a 24% decrease since the 1990s but the incidence remains ~10/100,000 and is increasing in the 25–35 years age group [2]. The peak age of CC is 30–35 years and 10-years survival is ~63%. In low-middle-income countries (LMICs) incidence and mortality is worse due to poor vaccination and late diagnosis. For locally advanced CC cases (stage IIB– IVA), chemo-radiotherapy (CRT) is the standard treatment with cisplatin (weekly 40 mg/m^2^ × 5). In IB3 and IIA2 stages, platinum-based CRT and brachytherapy is usually preferred [3]. However, ~28–64% are non-responders [4]. For recurrent and metastatic CC (clinical stage IVA-IVB) the treatment options are limited to palliative cisplatin-based chemotherapy with or without anti-vascular endothelial growth factor drug bevacizumab [3], with very poor survival (~5%). For the patients with renal insufficiency, cisplatin-based chemotherapy cannot be given as cisplatin is nephrotoxic [3].

Cisplatin and radiotherapy (RT) induce DNA damage to kill tumour cells [5,6]. Cisplatin causes both intra-and inter-strand cross-links [7] that stall replication forks in rapidly proliferating tumour cells. DNA damage activates signals to the cell cycle checkpoints (e.g., G1 checkpoint kinases ATM and CHK2, and intra S and G2/M checkpoint kinases ATR, CHK1 and WEE1) and promote DNA repair. HR-HPV inactivates p53 and pRb and thereby the G1/S checkpoint, making CC an ideal target for inhibition of intra-S and G2/M cell cycle checkpoints [8]. RT induces DNA single-strand breaks (SSBs) that are largely repaired by poly(ADP-ribose) polymerase1 (PARP1)-dependent mechanisms and double-strand breaks (DSBs) that signal ATR, CHK1 and WEE1 and are repaired by nonhomologous end-joining (NHEJ) and homologous recombination (HR) pathways [9]. Thus, inhibiting the DDR can result in chemo- and radiosensitisation [10]. PARP inhibitors (PARPi) were originally developed as chemo- and radiosensitisers [11] and 4 PARPi, including rucaparib (Rubraca^®^), are approved as single agents by the FDA for specific cancer types. Only eight trials of PARPi in CC are recorded on www.clinicaltrials.gov (accessed on 25 July 2022), including one in combination with cisplatin [12] and another with RT [13]. 

Cisplatin also causes acute kidney injury, even at the clinically recommended dose of 40 mg/m^2^, which is exacerbated in patients with large tumours (˃4 cm) that extended mainly at the pelvic side wall causing obstructive nephropathy resulting in discontinuation of the treatment [14]. Despite several pharmacological interventions being evaluated none have consistently ameliorated this injury [15]. There is evidence to suggest that cisplatin-induced kidney damage is associated with PARP activation and that PARP inhibition may reduce it [16,17]. We had previously shown that rucaparib reduced doxorubicin-induced cardiotoxicity [18] and therefore hypothesised that it would also be active against cisplatin-induced nephrotoxicity.

The poor response of advanced CC to the current CRT regimens along with toxicity issues means that any agents that can improve CRT response and/or reduce cisplatin-induced kidney injury are urgently needed. Therefore, the primary aim of this study was to evaluate the therapeutic potential of inhibitors of PARP, ATR, CHK1 and WEE1 as single agents and in combination with cisplatin and RT using a panel of CC cell lines. Additional aims focused on an investigation of possible determinants of sensitivity that have the potential to be predictive biomarkers and the potential of PARPi to ameliorate cisplatin-induced kidney injury. 

## 2. Materials and Methods

### 2.1. Chemicals and Reagents 

Unless stated otherwise, chemicals and reagents were purchased from Sigma-Aldrich (Poole, Dorset, UK). Cisplatin (cis-Diammineplatinum(II) dichloride) stock solution was prepared in sterile 0.9% *w*/*v* sodium chloride (NaCl) solution at 1 mM, filter sterilised through a 0.2 micron filter, aliquoted and stored at −20 °C. Rucaparib (PARP inhibitor, kind gift from Pfizer), VE-821 (ATR inhibitor), PF-477736 (CHK1 inhibitor) and MK-1775 (WEE1 inhibitor) (Selleckchem.com (accessed on 25 July 2022). Houston, TX, USA) were dissolved in dry DMSO at a concentration of 10 or 20 mM and stored in aliquots at −80 °C.

### 2.2. Cell Lines

Seven human cervical cancer cell lines were used: HeLa, SiHa, C33A, CaSki, ME-180, and HT-3. Cell lines were purchased from the American Type Culture Collection (ATCC) cell biology collection, authenticated by short tandem repeat (STR) profiling and used within 30 passages from purchase or authentication. They were confirmed as mycoplasma free (MycoAlert, Lonza, Basel, Switzerland). They were maintained in exponential growth in DMEM (HeLa, C33A SiHa and CaSki) or RPMI (ME-180 and HT-3) supplemented with 10% foetal calf serum at 37 °C in an atmosphere of 5% CO_2_ in air.

### 2.3. Cytotoxicity Assy

Exponentially growing cells were seeded at low density ranging from 50 to 1000 cells/well based on the plating efficiency (colonies formed/cells seeded) of the cell lines and allowed to adhere for 24 h. This was followed by treatment with different concentrations of drugs as single agent or in combination for 24 h. Ionising radiation/IR (X-irradiation; IR) was administered using a Gulmay Medical Xstrahl RS320 X-irradiator (Gulmay Medical, Chertsey, UK) at a rate of 3.15 Gy/min. The DMSO concentration was maintained at 0.5%. For chemo- and radio-sensitisation experiments, cells were incubated at 37 °C and allowed to adhere for 24 h. Cells were then either irradiated or exposed to cisplatin in the presence or absence of the inhibitors. Following 24 h of drug/IR treatment, the medium was replaced with drug-free medium, and cells incubated until colonies >30 cells formed. Cells were fixed in methanol: acetic acid (3:1 *v*/*v*) and stained with 0.4% crystal violet. Colonies were counted and the % survival for each treatment was calculated from the relative plating efficiency of treated versus vehicle (0.5% DMSO/only media) controls. LD50 and LC50 values defined as the dose (IR) or concentration (all drugs) where survival was 50% were measured by interpolation of the survival vs dose/concentration curves using GraphPad Prism software. The potentiation factor at 50% survival (PF50) is the ratio between survival at the LD50 or LC50 for IR or cisplatin alone and the LD50 or LC50 in combination with DDR inhibitor. Similarly, the potentiation factor at fixed dose/concentration i.e., 2 Gy ionising radiation (PF2-IR) and 0.3 µM cisplatin (PF0.3-cis) is the ratio of the survival at these doses/concentrations in the absence and presence of a DDR inhibitor. 

### 2.4. Measurement of Cellular Proteins and Their Phosphorylation by Western Blot

The expression of cell cycle checkpoint kinases and other DDR proteins was determined by Western blot. Activation of ATR, CHK1 and WEE1 and their inhibition were determined by measuring target phosphorylation (pCHK^S345^, pCHK1^S296^ and pCDK1^Y15^, respectively) by Western blot. Lysates were collected from exponentially growing cells with RIPA buffer containing 1% protease inhibitor cocktail (Thermo Fisher Scientific, Waltham, MA, USA), following manufacturer instructions. Protein content was estimated by Pierce BCA assay (Thermo Fisher Scientific) following manufacturers guidelines, samples were diluted to equal concentrations between 0.8 and 1 mg/mL in XT sample buffer and XT reducing agent (BioRad, Hercules, CA, USA), and boiled at 95 °C for 10 min. Lysates were separated by SDS-PAGE using 3–8% Tris-Acetate gels (BioRad), transferred to nitrocellulose Hybond^TM^ C membrane (Amersham, Buckinghamshire, UK), and blocked in 5% nonfat milk in tris buffered saline (TBS) pH 7.6 (Tris(hydroxymethyl)aminomethanol; sodium chloride, Fisher 7647-14-5; HCl, Fisher H/11/PB15) with 0.1% Tween 20 (TBS-T). Membranes were incubated with primary antibodies overnight at 4 °C. The following primary antibodies were used and diluted in 5% nonfat milk unless otherwise stated: α-tubulin (Sigma, T6074 1:80,000), ATM (Cell Signaling, 2873; 1:500 in 5% BSA), ATR (Santa Cruz, Sc515173; 1:200), CDK1 (Cell signalling, 9116 1:1000 in 5% BSA), CHK1 (Santa Cruz, Sc8408; 1:500), DNA-PKcs (Santa Cruz, Sc390849; 1:500), Ku70 (Abcam, Ab3114; 1:500), Ku80 (Abcam, Ab80592; 1:500), PARP1 (Biovision, 3001-100; 1:500), WEE1 (Santa-Cruz, 5285 1:500). ATR, CHK1 and WEE1 activity and inhibition was estimated by measuring pCHK1^S345^ (Cell Signalling, 2348 1:1000 in 5% BSA), pCHK1^S296^ (Cell Signalling 2349, 2348 1:1000 in 5% BSA) and pCDK1^Y15^ (Cell Signalling, 91111:1000 in 5% BSA). Next, the membranes were incubated with the horseradish peroxidase (HRP) conjugated secondary antibodies (Dako, P0447 and P0448; 1:2000) in 5% nonfat milk for 1 h at room temperature. For detection, Clarity Western enhanced chemiluminescence substrate (BioRad) was added and bands were visualised using the G-box gel documentation system (Syngene, Cambridge, UK), quantified using ImageJ software, and normalised to α-tubulin or Ponceau S stain. 

### 2.5. Immunofluorescence (IF) Microscopy

To measure PARP activation and co-localisation studies using IF, cells were plated on coverslips and allowed to adhere for at least 12 h prior to treatment with the IR, drugs/inhibitors. Optimisation of cisplatin concentration 500 uM used for the PARP activation study can be found in Appendix A. Cells were fixed with ice-cold methanol for at least 30 min before staining. Coverslips were washed in phosphate-buffered saline (PBS) containing 0.5% Triton X-100 (PBS-T) for 3 × 10 min. Washing was followed by blocking for 1 h with the blocking buffer (2% BSA, Sigma A2153; 10% (*w*/*v*) milk powder; 10% goat serum, Sigma G9023 in TBS-T) at room temperature. After the blocking step, primary antibodies were added and incubated at 4 °C overnight. The coverslips were subsequently washed 3 × 10 min in PBS-T followed by incubation with the secondary antibodies; Alexa Fluor™ 546/488 (1: 1000 dilution; Invitrogen) for 1 h at room temperature and then washed 3 × 10 min in PBS-T. Then, the coverslips were stained with DAPI (1: 1000; Sigma D9542) for 30 min at room temperature. Finally, the coverslips were mounted using anti-fade mountant (Invitrogen, prolong glass antifade mountant). Stained coverslips were observed under the Leica SPE confocal microscope and the images were analysed using ImageJ software. Primary antibodies used and their working dilutions were as follows. For detecting PARP-activity, Poly ADP-ribose (PAR) (E6F6A) Rabbit mAb (Cell Signaling, 83732) used at the dilution 1:1000, in blocking buffer. Cyclin E (HE12) (Santa cruz, sc-247) and RPA 32 kDa subunit (9H8) (Santa cruz, sc-56770) were used at the dilution 1:1000 in blocking buffer.

### 2.6. Cell Cycle Analysis

To determine the cell cycle effects of the ATR, CHK1 and WEE1 inhibitors exponentially growing cells were exposed to DMSO, 3 µM cisplatin or 3 µM cisplatin plus either 1 μM VE-821, 50 nM PF-477736 or 100 nM MK-1775 for 24 h. Cells were washed twice with PBS, with each washing collected to ensure no loss of cells, before being trypsinised and harvested. Following centrifugation at 1500 rpm for 5 min, the supernatant was discarded, and the remaining cell pellet was resuspended in 1 mL ice cold PBS and centrifuged (3000 rpm, 5 min). The supernatant was removed, and 1 mL of 70% ethanol was added dropwise to the cell pellet. Samples remained at 4 °C for a minimum of 1 h. Prior to staining, cells were washed twice in PBS to remove ethanol before eventually being resuspended in 800 μL PBS. RNase was added to cells (final concentration 1 mg/mL) alongside propidium iodide (PI) stain (final concentration 400 μg/mL). Cells were incubated in dark conditions at 37 °C for a minimum of 30 min. Cells were analysed for DNA content using a BD FACSCanto II flow cytometer. Data were stored and transferred to FCS Express 7 research edition^®^, De Novo software, Pasadena, CA 91107, USA for analysis. Doublets were excluded, and the sub-G1, G0/G1, S and G2/M populations were determined from the cell cycle histogram.

### 2.7. Determination of Acute Kidney Injury in Mice Treated with Cisplatin and Rucaparib

Female wild type CD-1 mice (Charles River Laboratory, Tranent, UK) were used in this study. All mice studies were conducted under Home Office licence PD86B3678 (Kevin Marchbank) at the Comparative Biology Centre (CBC), Newcastle.

The maximum tolerated dose of cisplatin in the 5-day acute kidney toxicity model was 10 mg/kg (Appendix A). In subsequent studies, mice were given cisplatin (10 mg/kg) via intraperitoneal (i.p.) injection. After 5 days, mice were terminally bled by cardiac puncture under anaesthesia, blood samples were collected in EDTA and kidney tissues were harvested in 10% buffered formalin solution. Plasma samples were analysed for urea and creatinine in the Newcastle Hospitals NHS laboratories. The whole kidney sections were analysed as described below, Section 2.8.

To determine the effect of PARP inhibition on cisplatin-induced acute kidney injury, mice received 10 mg/kg cisplatin i.p. on day 1 in combination with rucaparib at 1 mg/kg i.p. daily on days 1–5. In control groups, mice received only saline i.p. daily on days 1–5; 10 mg/kg cisplatin i.p. on day 1 and rucaparib at 10 mg/kg i.p. daily on days 1–5. The mice were humanely killed, and blood and tissues were collected as above. The combination study was done in two independent experiments, once with n = 3 mice per group and then n = 4 mice per group. The final data presented is a combination of the blocks.

### 2.8. Immunohistochemistry (IHC)

Formalin fixed paraffin embedded (FFPE) sections (4 µM) of the kidney on glass slides were dewaxed and hydrated using deionised water. Antigen retrieval was done using a pressure cooker in citrate buffer (pH 6) and to block endogenous peroxidase, slides were immersed in 3% H_2_O_2_ solution in TBS for 15 min. Slides were then washed in TBS-T for 2 × 10 min. Washing was followed by blocking (10% Normal Goat Serum, 1% BSA in TBS) for 1 h at room temperature. Slides were then washed with TBS-T 3 × 5 min before applying primary antibodies. After that, slides were washed with TBS-T 3 × 5 min. Next, the slides were incubated with secondary reagent (MenaPath^®^ Reagents) 30 min at room temperature followed by washing with TBS-T for 3 × 5 min and then stained with DAB (3,3’-Diaminobenzidine) for 3 min. Slides were then counterstained in haematoxylin. Primary antibodies used and their working dilutions used are as follows. For the PARP activity, PAR binding reagent MABE1016 (Millipore, MA, USA) was used at the dilution 1:1000 in TBS for 1 h at 4 °C. Lipocalin-2: anti-Lipocalin-2/NGAL affinity Purified Goat IgG (R&D, AF1857) used at 1: 1000 in TBS for overnight at 4 °C and with that, VisUCyte HRP Polymer Goat IgG Antibody (R&D, VC004-025) as the secondary agent was used. 4-HNE: Anti-4 Hydroxynonenal antibody (Abcam, ab46545) at 1: 1000 in TBS for overnight at 4 °C was used. For the PAR and 4-HNE staining, an additional step of Avidin-Biotin blocking (Vector laboratories, SP-2001) was done before applying the primary antibodies. Stained sections were scanned and quantified using Leica Aperio Image Scanner. IHC images were quantified following the H-score algorithm using the Aperio Image Scope software v11.2.0.780 (Leica Biosystems, Nussloch, Germany). H-score is defined as a multiplicative score of maximal stain intensity (0–3; where 0 was no staining, 1 was weak staining, 2 was moderate staining and 3 was strong staining), multiplied by the area of interest.

### 2.9. Statistical Analysis

GraphPad Prism 9, GraphPad software, San Diego, CA, USA was used for statistical analysis. *p* values < 0.05 were considered significant. Mean ± standard errors are shown in figures where applicable. * = *p* < 0.05, ** = *p* < 0.01, *** = *p* < 0.001, **** = *p* < 0.0001. 

## 3. Results

### 3.1. Cell Line Characteristics, Target Expression and Inhibition

A panel of 7 CC cell lines, which differed in their HPV status (Appendix A), growth rate and cloning efficiency (Appendix A) was investigated. DoTc2 was excluded from further study as they failed to form colonies for cytotoxicity assays. Then, from the remaining cell line panel, 4 CC cell lines (HeLa, SiHa, C33A, CaSki) were selected, as representing the diversity of HPV status in CC for more detailed in vitro evaluation (further justification is in the figure legend of Appendix A). The relative expression of the target proteins (PARP1, ATR, CHK1, WEE1) and additional potential determinants of sensitivity to PARP and cell cycle checkpoint inhibitors (ATM, DNA-PKcs, Ku-70 and Ku-80) were analysed (Figure 1A,B, Appendix A). The relative expression of the DDR proteins varied ~2 to 5-fold across the 4 CC cell lines (Figure 1B). Basal PARP activity was very low in all cell lines ~1.6–7.2 pmol PAR/10^6^ cells but total cellular PARP activity, measured in the presence of an activating oligonucleotide and excess NAD^+^ was 18–36 nmol PAR/10^6 cells (Appendix A), neither basal PAR nor PARP activation were related to PARP1 protein levels (Figure 1A,B). 

Both ionising radiation (IR) and cisplatin activated PARP (Figure 2A). PARP was activated in almost all cells immediately after irradiation, but cisplatin only activated PARP in ~20% of cells, which were subsequently identified as S-phase cells by co-localisation of PAR with RPA and Cyclin E (Figure 2B), and this was completely blocked by 1 µM rucaparib (Figure 2A). The VE-821, PF-477736 and MK-1775 inhibitors caused a concentration-dependent inhibition of phosphorylation of the targets of ATR, CHK1 and WEE1 (Figure 2C and Appendix A). Moreover, cisplatin activated these targets in all cell lines which were inhibited by their inhibitors (Figure 2D and Appendix A).

### 3.2. Single Agent Cytotoxicity, Chemo and Radiosensitisation

The cytotoxicity of the inhibitors alone is shown in Figure 3A–D and Appendix A. There was a fairly narrow range of sensitivity to rucaparib with C33A being most and HeLa least sensitive (2.5-fold difference in survival at 30 μM rucaparib, *p* = 0.0118). Interestingly, although VE-821, PF-477736 and MK-1775 all act at different points in the same cell cycle checkpoint pathway, the response of the cells to these drugs was different. HeLa cells were around 13x more sensitive to 30 μM VE-821 than SiHa cells (*p* = 0.0217), and SiHa cells were about 17x more resistant to 0.8 μM PF-477736 than CaSki and C33A (*p* = 0.0753) cells but CaSki cells were 15x more resistant to 1.6 μM MK-1774 than HeLa cells (*p* = 0.0577). 

The cells displayed a relatively narrow spectrum of sensitivity to both IR (LD50 in the range 1–2.5 Gy) and cisplatin (LC50 range 400–800 nM) (Appendix A). HeLa cells, which have high levels of DNA-PKcs, Ku70 and Ku80 (Figure 1B) was the most radio-resistant cell line, but it was also resistant to cisplatin and therefore this may merely reflect a greater resistance to DNA damage-induced cell death in this cell line.

Radiopotentiation was modest with all the inhibitors (Figure 3E–H and Appendix A). Rucaparib and VE-821 caused the greatest radiosensitisation on average (up to two-fold at 50% survival, Figure 3E,F,M). Cisplatin sensitisation was variable (Figure 3I–L and Appendix A), it was modest with rucaparib (<2-fold at 50% survival Figure 3I,N) but more substantial—up to six-fold—with VE-821 (Figure 3J,N). Chemosensitisation by PF-477736 and MK-1775 was more modest (up to 2-fold) across the panel (Figure 3K,L and Appendix A) and there seemed to be a general trend of VE-821>PF-47736=MK-17775. Chemo-radiosensitisation, i.e., sensitisation of increasing IR doses in the presence of 1 μM cisplatin by rucaparib (1 μM) was similar to radiosensitisation alone (Appendix A).

### 3.3. DNA Damage Induced Cell Cycle Effects and Impact of ATR, CHK1 and WEE1 Inhibitors

ATR, CHK1 and WEE1 inhibitors are thought to enhance cisplatin cytotoxicity largely through abrogation of S and G1 checkpoint signalling. In order to understand the hierarchy in terms of chemo- and radiosensitisation by VE-821, PF-477736 and MK-1775 we investigated their effect on cell cycle perturbations by cisplatin (Figure 4A, Appendix A). 

Cisplatin caused a very marked accumulation in S-phase but not G2. While VE-821, PF-477736 and MK-1775 alone had negligible effects on the cell cycle profile, they substantially reduced the S-phase accumulation in HeLa and SiHa cells and to a lesser extent in C33A and CASKi. Curiously, rather than also causing an attenuation of the G2 phase accumulation this phase was further markedly increased in HeLa and SiHa cells, but not C33A (Figure 4A). Overall, there was a significant relationship between target inhibition and reduction in S-phase for VE-821 but not PF-477736 and MK-1775 (Figure 4B). Interestingly, there was a positive relationship between ATR expression and VE-821′s impact on cell cycle changes but a negative relationship with CHK1 expression (Figure 4C). However, the cell cycle effects did not seem to be related to the chemosensitisation as, for example, VE-821 had a very modest effect on cisplatin-induced S-phase accumulation in C33A cells but caused the most profound chemosensitisation in this cell line.

### 3.4. Amelioration of Cisplatin-Induced Kidney Injury by Rucaparib

Despite the relatively modest effect of rucaparib on cisplatin cytotoxicity, there may still be a role for rucaparib in protecting from renal toxicity of cisplatin. To this end, we investigated the effect of rucaparib on cisplatin-induced acute kidney injury in mice. Cisplatin 10 mg/kg caused substantial increases in blood urea and creatinine that was reduced to approximately normal levels by co treatment with 1 mg/kg rucaparib (Figure 5A). 

Histological analysis using periodic acid Schiffs (PAS) staining revealed extensive structural damage at the kidney’s proximal tubular regions, characterised by tubular cast formation, in response to cisplatin (Appendix A). Co-treatment with rucaparib caused a marked decrease in cisplatin-induced tubular cast formation (Appendix A). Lipocalin-2 expression (a biomarker of kidney injury) was significantly elevated in response to cisplatin, which was markedly ameliorated by rucaparib (Figure 5B). In depth spatial quantification at the kidney’s cortex and medulla regions were also done. Cisplatin increased lipocalin-2 expression to the greatest extent in the kidney’s cortex regions (cortex: ~35 fold vs. medulla: ~5.5 fold) (Figure 5B).

To determine the reason for the damage being more focussed in the cortex and how rucaparib was preventing this, we looked for evidence of oxidative stress and PARP activation. We found that cisplatin caused an increase in 4-HNE levels across the entire kidney both at the cortex (~6.5 fold) and at the medulla regions (~4.5 fold) (Figure 5C).

Cisplatin also caused an increase in PAR formation, which, in contrast to 4-HNE was much greater in the cortex regions (~9.0 fold) but only ~4-fold in the medulla (Figure 5D). Thus, the area of maximum PARP activation corresponded to that of maximum injury (Figure 5B). Rucaparib had no significant effect on the level of 4-HNE but, both PAR formation and lipocalin-2 expression was inhibited by rucaparib. Since PAR levels but not 4-HNE correlate with the pattern of lipocalin expression, it seems likely that the toxicity in the cortex is due to PARP activation.

## 4. Discussion

The results described in this study are the first direct comparison of the cytotoxicity of PARP, ATR, CHK1 and WEE1 inhibitors as single agents and in combination with cisplatin and radiotherapy. There are very limited clinical studies of DDR inhibitors in cervical cancer(CC) to date. PARPi are being studied in combination with platinum, RT [13] in early phase trials and outcomes are awaited. Cell cycle checkpoint kinase inhibitors are being explored clinically in gynaecological cancer including the ATR inhibitor, AZD6738 [19] and the WEE1 inhibitor AZD1775 (formerly MK-1775) [20] is currently being evaluated in combination with cisplatin and radiotherapy in cervical, and other cancers in a Phase 1 trial [20]. Our aim was to evaluate these DDR inhibitors in CC cells to inform future clinical studies of the relative potential of these drugs in CC both as single agents and as chemo- and radiosensitisers, to investigate the mechanisms underpinning their action and to identify potential determinants of sensitivity.

Having established that the cells expressed the target enzymes, and other DDR proteins that have been previously suggested as determinants of sensitivity to the DDR inhibitors, and that the inhibitors were active against their targets, we investigated their cytotoxicity as single agents. All cells showed concentration-dependent decreases in survival in response to all the inhibitors, but they differed in their rank order. The rank order of sensitivity would be expected to be different between rucaparib and the cell cycle checkpoint kinase inhibitors. However, the rank order to VE-821, PF-477736 and MK-1775 also differed. This suggests either (i) that there are subtle differences in response to targeting different points in the same pathway or (ii) that the balance of the different DDR pathways between the cell lines dictated their differing responses inhibition of different points in the pathway. The sensitivity to the S/G2/M checkpoint inhibitors is hypothesized to be greatest in cells lacking G1 control through p53/pRb dysfunction [21]. HPV infection inactivates both p53 and pRb function. We have observed in C33a, the levels of the p53 protein was very high, as expected (HPV negative, p53 mutated). In HeLa, SiHa and CaSki, the presence of the HPV leads to p53 degradation and therefore low p53 levels (Appendix A). Interestingly, the two cell lines that were not associated with HPV (C33a and HT-3) had pathological mutations in p53 and pRB suggesting that inactivation of these two proteins is a prerequisite for the development of cervical cancer. It was therefore not possible to determine if the dysfunction of the p53-pRb pathway contributed to the cytotoxicity of VE-821, PF-477736 or MK-1775. There was no relationship between the cytotoxicity (LC50) of rucaparib, VE-821, PF-477736 or MK-1775 (Figure 3A–D) and the expressions of their target proteins (Figure 1B) in this cell line panel. ATM mutation is well documented as a determinant of sensitivity to ATR [10,22] and a recent study showed that low ATM protein levels were associated with ATR inhibitor cytotoxicity in a panel of neuroblastoma cell lines [23]. Indeed, in many ongoing clinical studies of ATR inhibitors, ATM mutation and/or expression is being used for patient selection [24,25,26,27,28]. In this panel of CC cells, low levels of ATM did appear to be associated with greater VE-821 cytotoxicity (Appendix A) but not PF-477736 or MK-1775 sensitivity. However, in contrast to previous reports indicating that high DNA-PKcs levels conferred sensitivity to ATR inhibition in matched cell lines [29], our data suggest this might not be universally applicable as high DNA-PKcs was related to resistance to VE-821 cytotoxicity (i.e., SiHa, Figure 1B and Figure 3B). Previous studies suggest DNA-PKcs and ATR expression are positively correlated in glioma models [29], which might suggest that the balance of the two pathways is important for cell viability. Within our limited panel there was a correlation between ATR and DNA-PKcs protein levels (Figure 1B) but our analysis of TCGA data indicates a poor correlation in mRNA expression in cervical cancer (Appendix A). Further studies are needed to validate whether DNA-PKcs protein, but not mRNA, expression is a suitable additional biomarker of ATR inhibitor sensitivity.

DDR inhibitors are also under clinical evaluation in combinations studies with both conventional cytotoxic and molecularly targeted agents [10]. Such studies need careful evaluation of appropriate dose and schedule as they are less likely to exploit tumour-specific dysregulation of the DDR by synthetic lethality and may only serve to increase toxicity in parallel with anti-tumour activity. Radiotherapy, as it is tumour directed, particularly with advanced radiotherapy technologies (e.g., SABR hypofractionation, proton beam) may be the safer combination. However, radiosensitisation was fairly modest, with rucaparib being the most potent radiosensitiser (Figure 3M). Interestingly, the least radiosensitisation was observed in the most radiosensitive (C33A) and the greatest radiosensitisation was seen in the cells that were most radioresistant (HeLa) suggesting that PARP has a role in their response to radiation. However, this was not related to PARP1 protein levels as C33A had the highest level. It should be noted here that PARP activity is only poorly correlated with PARP1 protein levels [30] and therefore potential biomarkers should be based on PARP activity rather than protein level. The cell cycle checkpoint kinase inhibitors were less potent radiosensitiser and in general VE-821 was more potent than PF-477736 or MK-1775. The greatest radiosensitisation was observed at higher radiation doses, as has been previously observed [31], suggesting a greater therapeutic potential in combination with hypofractionated radiotherapy. 

High concentrations of cisplatin were needed to activate PARP to a lesser degree than ionising radiation, and rucaparib caused a very modest increase in cisplatin cytotoxicity (Figure 3N) suggesting that PARP may have a minor role in cisplatin toxicity. Published data for the enhancement of platinum agent cytotoxicity/antitumour activity in the pre-clinical setting is somewhat variable reviewed in [32] and may be related to the homologous recombination repair (HRR) status of the cells, as defects in this pathway confer sensitivity to both platinum agents and PARP inhibitors [33]. Another contributor to the modest nature of chemosensitision by rucaparib is that cisplatin only activated PARP in about 20% of the cells, which were identified as those going through S-phase (Figure 2A) when cisplatin-DNA cross-links are converted to strand breaks. VE-821 was the most potent chemosensitiser (Figure 3N) of all four inhibitors. In general, across the panel VE-821 caused the greatest chemosensitisation and MK-1775 the least. All inhibitors caused a similar attenuation of cisplatin-induced S-phase accumulation within individual cell lines, but this varied between cell line (i.e., attenuation was greatest with all inhibitors in HeLa and SiHa and least in C33A and CaSki). There was a significant relationship between target inhibition and reduction in S-phase for VE-821 (Figure 4B) as well as a positive relationship between ATR expression and VE-821′s impact on cell cycle changes (Figure 4C). But there were no similar correlations for PF-477736 or MK-1775. The impact of the inhibitors on the cell cycle distribution after cisplatin did not correlate with their different chemosensitisation potency or the different extent of chemosensitisation between the cell lines. These data suggest that cisplatin sensitisation is unlikely to be directly attributable to cell cycle checkpoint abrogation. Instead, it is possible that it is due to the inhibition of HRR through a coupling of ATR-mediated checkpoint activation and HRR [10]. Our recent studies have also shown that VE-821, PF-477736 and MK-1775 all block HRR in cervical and ovarian cancer cell lines and this is the mechanism underlying the synergy with PARP inhibitors [34]. 

We recognize the limitations of in vitro studies in predicting clinical outcome, and indeed in vivo xenograft studies also do not necessarily translate well clinically [35]. However, studies on the individual inhibitors in combination with cisplatin and (or) ionizing radiation have been conducted previously [12,36,37,38,39,40,41] and clinical trials of such combinations are underway. The promising data described here justify that further follow-up studies using patient-derived CC xenografts are required. Although, developing orthotopic xenograft model of CC is technically very challenging [42,43]. Despite the modest chemosensitisation by rucaparib there may still be a role for rucaparib clinically through renal protection. We report here the first study of an FDA approved PARPi i.e., rucaparib, to test amelioration of cisplatin-induced acute kidney injury in the pre-clinical setting. Previous studies by pharmacokinetic analysis in CD-1 mice revealed distribution of cisplatin in the kidney is higher compared to any other organs [44]. Bioaccumulation of cisplatin causes biotransformation into toxic metabolite (thiol-platinum compound/nephrotoxins) that increases oxidative stress which was observed in our study by 4-HNE analysis (Figure 5C). Cisplatin-induced oxidative stress is thought to cause DNA breaks that activate PARP [45]. It has been reported that cisplatin accumulation is greater in the cortex where most damage is observed [46], but we found that 4-HNE was increased by cisplatin across the whole kidney sections (both in the cortex and in the medulla). Importantly, both PARP activation (Figure 5D) and injury (Figure 5B) were seen predominantly in the cortex. Cisplatin has been shown to severely reduce NAD^+^ levels in the kidney [47]. NAD^+^ is the substrate for PARP, and PARP1 is the major consumer of NAD^+^ in response to base oxidation and DNA single strand breaks caused by oxidative stress. Therefore, PARP activation is likely to be the principal cause of the NAD^+^ depletion and injury. The DDR has previously been suggested as a mediator of cisplatin-induced nephrotoxicity with the potential for the use of DDR inhibitors as protective agents; however this study did not consider PARPi [47]. Importantly, rucaparib blocked both the PARP activity and the injury but had no impact on the 4-HNE. These data demonstrate clearly that cisplatin-induced acute kidney injury is due to PARP activation rather than oxidative stress per se. Whether it is the reduction of NAD^+^ [48] or the abundance of PAR that triggers a specific cell death pathway involving AIF (Apoptosis-inducing factor) known as “parthanatos” [49] has not been determined, but whatever the pathway it is blocked by PARP inhibition and the viability of the kidney is maintained.

## 5. Conclusions

The data reported here, which are the first to compare DDR inhibitors that are either clinically approved (rucaparib), have entered clinical trial (PF-477736 and MK-1775) or a close analogue of a drug in clinical trial (VE-821 is related to Berzosertib). We show that all have single agent activity in cervical cancer cell lines and can increase the cytotoxicity of the current standard of care. ATR, CHK1 and WEE1 inhibitors show the greatest promise in combination with cisplatin. PARPi may cause the greatest radiosensitisation effect but their clinical potential may be greatest in combination with cisplatin because we show conclusively that cisplatin-induced kidney damage is due to PARP activation and that PARP inhibition can ameliorate kidney injury, which is dose limiting for use of cisplatin.

## Figures and Tables

**Figure 1 cancers-14-04288-f001:**
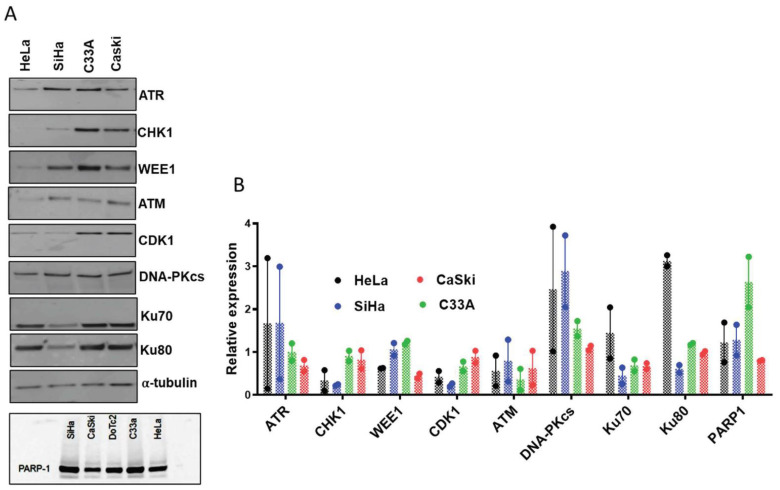
DDR protein expression in cervical cancer cell lines: (**A**) representative Western blots of key DDR proteins in cell lysates prepared from exponentially growing cells. (**B**) Densitometric analysis of replicate Western blots as shown in (**A**). Data are mean of two independent experiments. Protein expression was normalised to the loading control, α-tubulin or Ponceau S staining (for PARP-1) of the entire blot as given in Appendix A. Although, the PARP-1 expression in the DoTc2 was checked together with four other cell lines in the Western blot but, due to the very low clonogenic efficiency this cell line was excluded from all the studies hereafter and also excluded from the densitometric analysis in (**B**).

**Figure 2 cancers-14-04288-f002:**
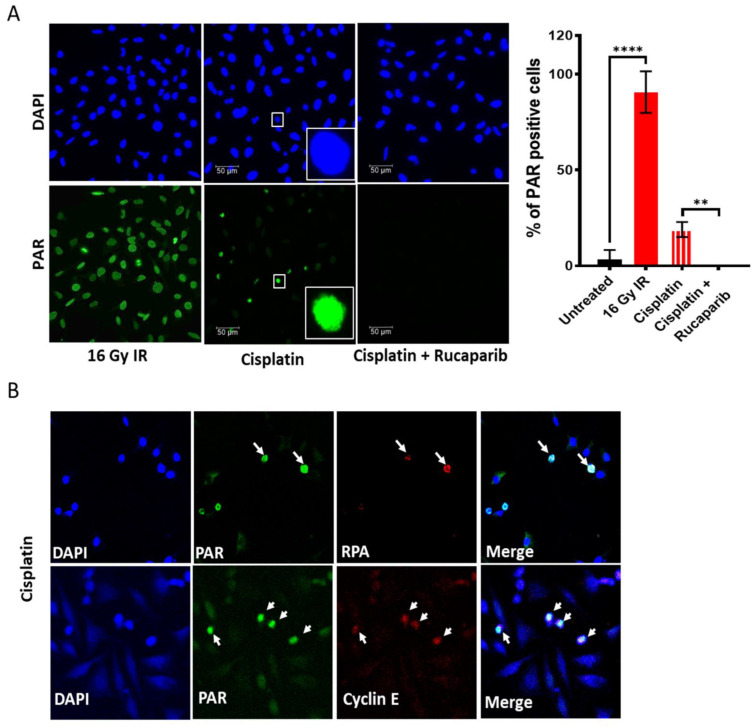
DDR activation by ionising radiation (IR) and cisplatin and inhibition by drugs targeting these pathways: PARP activity was measured by immunofluorescence detection of the product, PAR, in exponentially growing HeLa cells. Cells were fixed immediately after exposure to 16 Gy IR or 6 h incubation with 500 uM cisplatin at 37 °C. (**A**) Representative images are shown alongside a bar chart of the percentage of PAR positive nuclei, data are mean ± SD from four independent experiments. The difference of PAR positive cells between untreated, IR and cisplatin ± rucaparib treated groups were compared using unpaired *t*-test. ** and **** are *p* < 0.01 and 0.0001 respectively. (**B**) Co-localisation of PAR positive HeLa cells with replication stress marker RPA and cell proliferation marker Cyclin E indicates PARP activation in replicating cells only in response to cisplatin. Images at 40× magnification. (**C**) pCHK1S345, pCHK1S296, and pCDK1Y15 levels in cells treated with increasing concentrations of (**i**) VE-821, (**ii**) PF-477736 and (**iii**) MK-1775, respectively (Appendix A). Data represents mean ± SEM (N = 3). (**D**) Analysis of ATR, CHK1 and WEE1 activity from densitometric analysis of pCHK1S345, pCHK1S296, and pCDK1Y15, respectively, in Western blots of exponentially growing cell exposed to 0.5% DMSO alone, 3 μM cisplatin and 3 μM cisplatin + 1 μM VE-821 (VE), 50 nM PF-477736 (PF) or 100 nM MK-1775 (MK) for 24 h prior to harvest cells and lysate preparation (Appendix A). Data represents mean of two independent experiments.

**Figure 3 cancers-14-04288-f003:**
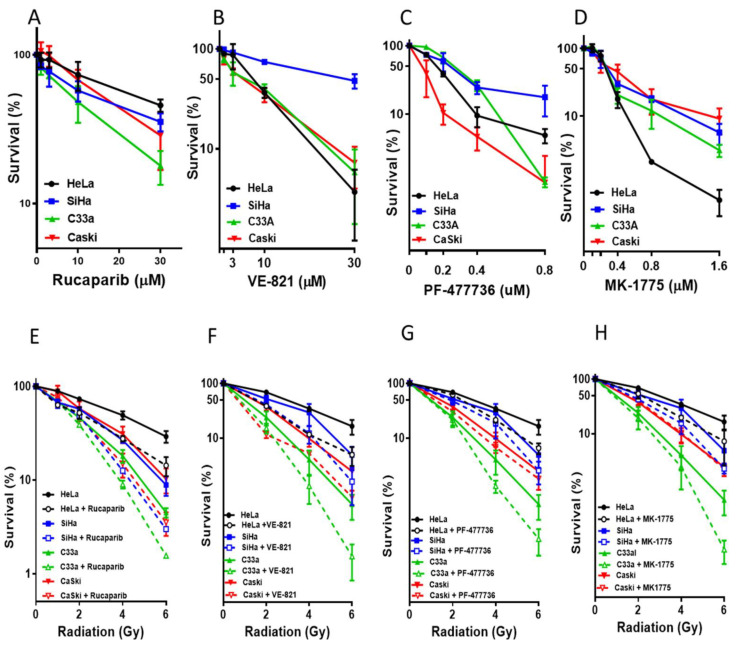
Single-agent cytotoxicity, radiosensitisation and chemosensitisation by the DDR inhibitors: Survival of cervical cancer cells following exposure to single agent (**A**) rucaparib, (**B**) VE-821, (**C**) PF-477736) and (**D**) MK-1775. Cells were exposed to increasing concentrations of the DDR inhibitors for 24 h and survival was determined by clonogenic assay. Survival is given as a percentage relative to the survival in 0.5% DMSO only as the control. Data are mean ± SEM from three independent experiments. Survival of cervical cancer cell lines exposed to increasing doses of ionising radiation (IR) ± (**E**) 1 μM rucaparib, (**F**) 1 μM VE-821, (**G**) 50 nM PF-477736, or (**H**) 100 nM MK-1775. Cells were exposed for 24 h and survival was determined by clonogenic assay. Survival is normalised to vehicle (DMSO 0.5%) or inhibitor alone. Data represent means ± SEM from three independent experiments. Survival of cervical cancer cell lines exposed to increasing doses of cisplatin ± (**I**) 1 μM rucaparib, (**J**) 1 μM VE-821, (**K**) 50 nM PF-477736, or (**L**)100 nM MK-1775. Cells were exposed for 24 h and survival was determined by clonogenic assay. Survival is normalised to vehicle (DMSO 0.5%) or inhibitor alone. Data represents means ± SEM from three independent experiments. (**M**) The heatmap represents the radiosensitsation potency of the inhibitors, which is based on the PF50 (Appendix A) with the PF50 values given within each cells of the heatmap. (**N**) The heatmap represents the chemosensitsation potency of the inhibitors, which is based on the PF50 (Appendix A) with the PF50 values given within each cells of the heatmap.

**Figure 4 cancers-14-04288-f004:**
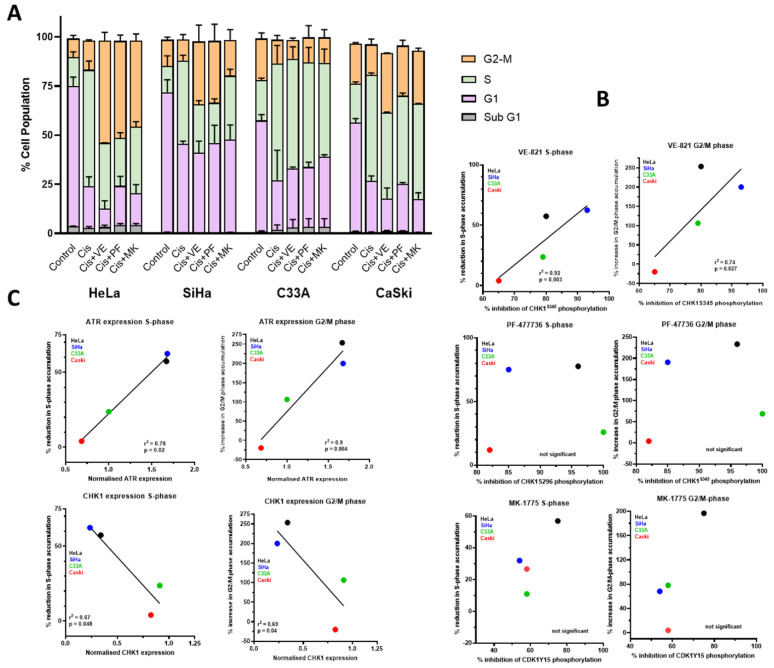
Effect of VE-821, PF-477736 and MK-1775 on the cell cycle control and its relation to target expression and inhibition: (**A**) Cell cycle profiles of exponentially growing cells were treated with vehicle control (0.5% DMSO), or 3 μM cisplatin (Cis) ±1 μM VE-821 (VE), 50 nM PF-477736 (PF) or 100 nM MK-1775 (MK) for 24 h before DNA content was analysed. At least 2000 events per sample were recorded. Data are the mean and range of values from two independent experiments. (**B**) Changes in cisplatin-induced S and G2/M accumulation vs. extent of inhibition cisplatin-induced activation of ATR by 1 μM VE-821, of CHK1 by 50 nM PF-477736 or of WEE1 by 100 nM MK-1775. Data are taken from Figure 2C and Figure 4A. (**C**) Changes in cisplatin-induced S and G2/M accumulation vs baseline ATR or CHK1 expression in the cell lines. Data are taken from Figure 1B and Figure 4A.

**Figure 5 cancers-14-04288-f005:**
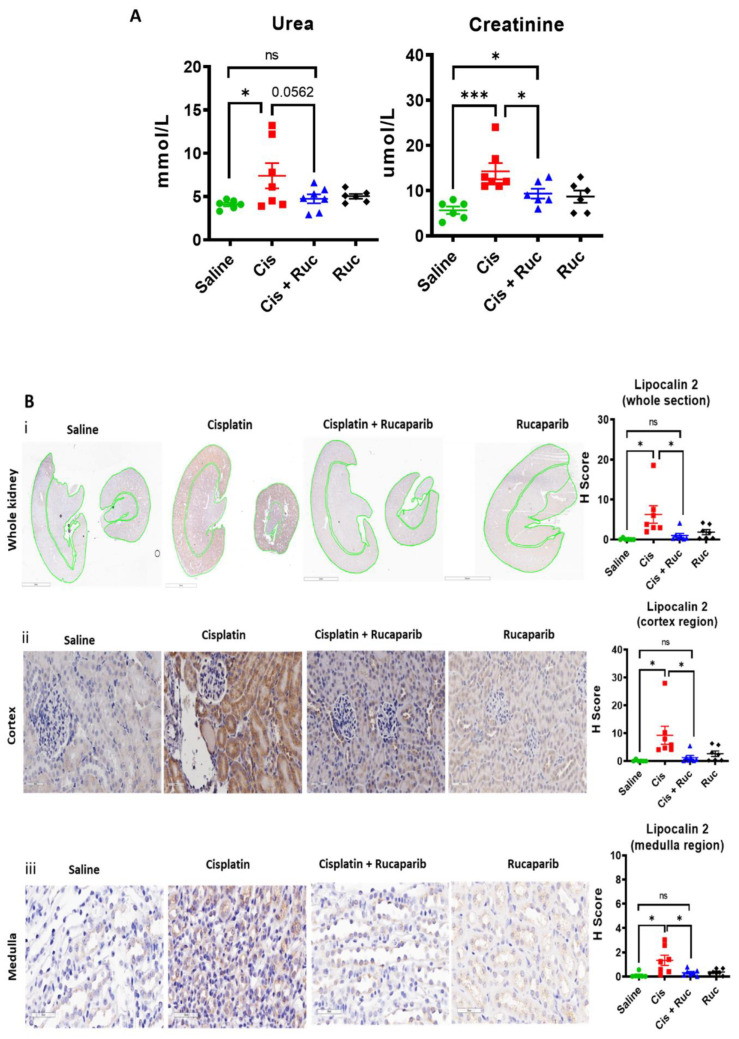
Cisplatin-induced-acute kidney injury is associated with PARP activation and is prevented by rucaparib: Mice were treated with a single dose of 10 mg/kg cisplatin (Cis) on day 1 with or without 5 daily doses of 1 mg/kg rucaparib (Ruc). At the end point (day 5), mice were humanly killed, and blood and kidney tissues were harvested for analysis. (**A**) Blood urea and creatinine levels. Each data point is from a single mouse (N =≥ 6 mice per group). The difference of urea and creatinine levels between saline and cisplatin ± rucaparib treated groups were compared using unpaired *t*-test, ns = nonsignificant, * and *** are *p* < 0.05 and 0.001 respectively. (**B**) Representative images of kidney toxicity biomarker lipocalin- 2 expression. IHC images were scanned under the Aperio image scanner and staining intensities (H-score) were quantified for the sections as a whole, cortex and medulla regions separately. (**i**) Whole kidney sections (0.9× magnification, 3 mm scale bar and the green lines are drawn to separate cortex and medulla regions for quantification), (**ii**) cortex and (**iii**) medulla regions (40× magnification, 50 µM scale bar). Each data point is from a single mouse (N =≥ 6 mice per group). The difference of lipocalin-2 expression between saline and cisplatin ± rucaparib treated groups were compared using unpaired *t*-test, ns = nonsignificant and * is *p* < 0.05. (**C**) Representative images of oxidative stress biomarker 4-HNE expression. IHC images were scanned under the Aperio image scanner and staining intensities (H-score) were quantified for the sections as a whole, cortex and medulla regions separately. (**i**) Whole kidney sections (0.9× magnification, 3 mm scale bar and the green lines are drawn to separate cortex and medulla regions for quantification), (**ii**) cortex and (**iii**) medulla regions (40× magnification, 50 µM scale bar). Each data point is from a single mouse (N =≥ 6 mice per group). The difference of 4-HNE expression between saline and cisplatin ± rucaparib treated groups were compared using unpaired *t*-test, ns = nonsignificant, * and ** are *p* < 0.05 and 0.01. (**D**) Representative images of PARP activation, measured by PAR expression (PAR positive cells are indicated with arrows). IHC images were scanned under the Aperio image scanner and staining intensities (H-score) were quantified for the sections as a whole, cortex and medulla regions separately. (**i**) whole kidney sections (0.9× magnification, 3 mm scale bar and the green lines are drawn to separate cortex and medulla regions for quantification), (**ii**) cortex and (**iii**) medulla regions (40× magnification, 50 µM scale bar). Each data point is from a single mouse (N =≥ 6 mice per group). The difference of PAR expression between saline and cisplatin ± rucaparib treated groups were compared using unpaired *t*-test, ns = nonsignificant, ** and *** are *p* < 0.01and 0.001.

## Data Availability

The data presented in this study are available in this article and Appendix A.

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
