# Peer review of "Determining the Potential of DNA Damage Response (DDR) Inhibitors in Cervical Cancer Therapy"

_cancers, 2022, doi:10.3390/cancers14174288_

Round 1

Reviewer 1 Report

The study at hand compares 4 DDR inhibitors and examines a mouse model to reduce kidney damage of cisplatin treatment. The paper is of high interest, but study design and quality of presentation need major improvements. The study needs major revision including additional experiments and extensive reorganization of data. My detailed critique can be found below:

·        Figure1: Needs a control cell line, like primary or immortalized Keratinocytes (for example https://www.atcc.org/products/pcs-200-010)

·        Please optimize the positioning of the CHK1, WEE1, ATM, Ku70 WB-bands in Figure 1A. The bands should be approximately in the middle of the box.

·        Why is Parp-1 separate and with an additional cell line? If it’s nowhere else in the paper, please exclude it from the representative image as well.

·        Figure 1B: Relative expression. What was the expression normalized to? It should be normalized to the control Keratinocyte cell line. Please conduct a 3rd repeat and include appropriate statistics. The current densitometry appears to strongly vary from the representative images in many cases. ATR: HeLa is lowest in image, WEE1: C33a is at least twice as strong as SiHa and CaSki in image, ATM: CaSki is strongest in image, Ku70 and 80: HeLa, C33a and CaSki appear approximately the same in image. Please create an individual graph for each protein.

·        Figure 2A: 500µM is a very high dose for HeLa treated with cisplatin. In fact it’s approximately 500 times higher than the LC50 you report in the supplements. As this concentration is posed to wipe out the HeLa population it is not suitable for determining PARP-response via IF. On that same note, SiHa are notoriously cisplatin resistant, so I don’t think the SiHa LC50 reported in the supplements is accurate.

·        Figure 2C: Please repeat a 3rd time and include relevant statistics. Also include the representative western blot images in the main figure. I recommend splitting figure 2 after panels A and B. As an aside, ATR pathway activation can be measured directly via p-ATR.

·        Figure 2C and D aspect ratios have been altered, please fix (letters look compressed in their width).d

·        Figure 2D: please show the entire dose series western blot (all inhibitor doses) in figure 2 to allow direct comparison.

·        Figure 2D Why do some cell lines have no data for the highest dose?

·        Figures 2C and D have graph titles, where previous graphs don’t. Figure 2D’s graph title sits below the graphs. Please aim for consistency

·        Mainly my problem is that SF4 has a very busy layout that makes it extremely difficult to decipher what is going on. I suggest eliminating the cell lines that aren’t in the main manuscript and reorganize the layout to make the figure much clearer. The representative western blots should be shown with the corresponding densitometry in the main figure. SF4 also has nothing to do with figure 2D, as only one dose of each respective inhibitor is shown.

·        Figure 3A-L: why does each graph get a panel letter, but Figure 2D is 4 graphs in one panel? Be consistent.

·        Figure 3A-D: Please include relevant statistics when you describe which cell lines were more sensitive than others. All these experiments need a Keratinocyte control cell line included (see 1st point of critique)

·        Restructure the heatmaps so they fit side by side.

·        Were any of these sensitizations significant and synergistic, or merely additive? There is a continuous lack of statistical analysis. Please revise the entire manuscript and include analysis beyond descriptive statistics (Quantification in Figure 2A is an example of what all graphs should look like.

·        Figure 4A: The original flow cytometer histograms with gates need to be shown in the supplements. Please also include graphs to demonstrate all relevant gating strategies.

·        4B and C: what do the regression lines mean? Why would a regression across cell lines be appropriate? Please significantly expand on the interpretation of this data.

·        Figure 5: Please include data on tumor volume to allow the comparison of tumor volumes between cisplatin and cisplatin&rucaparib

Author Response

We thank the reviewer for considering the work in this paper of high interest. We also thank the reviewer for the insight and helpful comments to enrich the quality of our manuscript. We have addressed these critique point by point and also included the amendments in the revised manuscript for publication in this journal. Please see below the answers to the comments and we hope, the reviewer is now confident with our honest clarification.

Reviewer 2 Report

While the authors have systematically demonstrated how various individual drugs can improve the overall efficacy of CRT, as well as work as single agent anti-cancer drugs, the manuscript would gain a lot by including PDX models and/or cancer models as opposed to just CD-1 mice model. While the in-vitro data is compelling, often in-vitro does not translate into invivo potency and hence the authors need to either support their data with invivo models or include possible explanation or limitations to their study and interpretation in the discussion. Also, the authors should mention why high dose of cisplatin (500uM) is used for their initial PAR experiment when dose of 10uM Cisplatin and higher often causes cytotoxicity and a great amount of DNA damage to the cells; this does not necessarily correlate with author's own data with PAR induction (Fig 2A-B)

Author Response

We thank the reviewer for the helpful comments which we address below.

Round 2

Reviewer 1 Report

I thank the authors for adressing my criticisms as much as possible and hope their work finds many interested readers!